# The validation of the Beijing version of the Montreal Cognitive Assessment in Chinese patients undergoing hemodialysis

Ru Tian[1], Yidan Guo[1], Pengpeng Ye[2], Chunxia Zhang[1], Yang Luo[1]*

**1** Division of Nephrology, Beijing Shijitan Hospital, Capital Medical University, Bejing, China, **2** Division of Injury Prevention and Mental Health National Center for Chronic and Non-communicable Disease Control and Prevention, Chinese Center for Disease Control and Prevention, Beijing, China

* luoyang96@163.com

## Abstract

### Objective

Cognitive impairment is common among hemodialysis patient, but still lack adequate screening in clinical settings. The Montreal Cognitive Assessment (MoCA) is reportedly to be a sensitive screening tool for cognitive impairment, but its clinical value in patients undergoing hemodialysis is not well established. We aimed to validate the utility of the Beijing version of the MoCA (MoCA-BJ) for detecting cognitive impairment in comparison to a detailed neuropsychological battery as the gold standard.

### Methods

We assessed 613 patients undergoing hemodialysis using the MoCA-BJ, the Mini-Mental State Examination (MMSE), and a comprehensive neuropsychological battery. Cognitive dysfunction was defined by the fifth version of the Diagnostic and Statistical Manual of Mental Disorders (DSM-V). Spearman's correlation and linear regression were used to estimate the performance of the MoCA-BJ and MMSE in predicting cognitive impairment. A receiver operating characteristic (ROC) curve analysis was used to evaluate the utility of various cut-offs of the MoCA-BJ and MMSE for predicting cognitive impairment.

### Results

Cognitive impairment was diagnosed in 80.91% (496/613), 75.69% (464/613), and 61.34% (376/613) of the patients using the DSM-V, MoCA-BJ, and MMSE, respectively. Spearman's rank correlation analysis indicated that the MoCA-BJ was significantly correlated with the neuropsychological battery ($r_s$ = 0.639, p<0.001), whereas the MMSE had a weaker correlation with the battery. The area under the ROC curve for cognitive impairment diagnosis using the MoCA-BJ was 0.891 (95% confidence interval: 0.859–0.924) while using the MMSE was 0.823 (95% confidence interval: 0.786–0.860). The optimal MoCA-BJ cutoff score in discriminating patients with and without cognitive impairment was 24 points with a sensitivity of 0.877 and specificity of 0.752.

**Data Availability Statement:** Data are available and uploaded as supplementary files.

**Funding:** This project is supported by a grant from the Beijing Municipal Science & Technology

Commission, NO. Z161100002616005 (https://mis.kw.beijing.gov.cn/out/typt/staticHTML/homePage.html) to YL. The funder had no role in study design, data collection and analysis, decision to publish, or preparation of the manuscript.

**Competing interests:** The authors have declared that no competing interests exist.

## Conclusion

The MoCA-BJ offers good sensitivity and specificity levels in detecting cognitive impairment in hemodialysis patients. These findings support the utility of the MoCA-BJ as a screening tool for cognitive impairment in Chinese patients undergoing hemodialysis.

## Introduction

The prevalence of cognitive impairment in patients undergoing hemodialysis (HD) is approximately 30–80%, which is two to four-fold higher as much as that in the general population [1–3]. In addition to being associated with the occurrence of cerebrovascular diseases and other adverse clinical outcomes, cognitive impairment may also influence adherence to medication and dietary management in hemodialysis patients [4, 5]. Under such circumstances, it seems urgent to find a rapid yet sensitive tool in the early detection of cognitive impairment among those patients.

Although comprehensive neuropsychological testing remains the gold standard for assessing cognitive function, it is not always logistically or economically feasible in clinical settings, the Montreal Cognitive Assessment (MoCA) is a brief screening test for cognitive impairment that covers major cognitive domains including memory, language, attention, orientation, visuospatial ability, and executive functions[6]. Several validation studies regarding the discrimination of different degrees of cognitive impairment were published in the last 10 years, they showed great variability in terms of sample sizes, normative scores and cut-off points for detecting people with Mild Cognitive Impairment (MCI) and Alzheimer's disease (AD) [7, 8]. Bosco A et al [9]compared a group of people with probable Alzheimer's Disease (AD) with healthy counterparts, the results showed that the optimal cutoff for a diagnosis of probable AD was a MoCA less than 14 in those Italian population. The Beijing version of MoCA (MoCA-BJ) has been widely used in mainland China for the screening of cognitive function [10]. Huang L et al [11] validated the Chinese version of MoCA and found that the MoCA was an effective cognitive test to distinguish between NC, MCI, mild and moderate AD among the Chinese elderly with various levels of education. Other studies have assessed the ability of the MoCA in the detection of cognitive impairment in a variety of diseases including Alzheimer's disease, cerebral small vessel disease, and stroke [7, 12].

Data concerning the predictive value of MoCA in hemodialysis patients were mostly from other ethnic groups and the sample size of the previous studies was relatively small [13, 14]. Therefore, we aimed to validate the sensitivity, specificity and predictive value of the MoCA-BJ as a screening tool in detecting cognitive impairment which was determined by a comprehensive neuropsychological battery in a group of Chinese hemodialysis patients.

## Methods

### Participants

The study was performed using the data repository of the cohort study of CI in Chinese patients undergoing hemodialysis (CODE) (ClinicalTrials.gov ID: NCT03251573), which including 613 patients recruited from 11 HD centers in Beijing between April 2017 and June 2017. The mean age of participants was 63.7±7.8 years old, female participants were 258 (42.1%), the ratio of education level over 12 years was 72.4%. The eligibility criteria for participants were as follows: (1) aged 50–80 years, (2) had end-stage kidney disease, (3) were treated

with long-term outpatient HD for the previous ≥3 months, (4) their HD team agreed to join the investigation and patients were willing to provide written informed consent, (5) ability to complete a 90 min cognitive and physical function battery, and (6) first language is Chinese.

The exclusion criteria for all participants were as follows: (1) were unable to participate for reasons such as sensory (e.g., visual and hearing) or motor impairment, (2) had a life expectancy of <6 months according to the evaluation from the physicians, (3) experienced disturbance of consciousness or were recently diagnosed with psychosis, and (4) had a planned kidney transplantation within 6 months of baseline.

At the enrollment, sociodemographic information, clinical history, and HD vintage were obtained by participant reports and patients' electronic or paper charts. Pre-dialysis blood tests included measurement of the serum levels of hemoglobin, albumin, calcium, phosphate, and intact parathyroid hormone; the single-pool Kt/V were obtained from all the subjects.

## Neuropsychological assessment

Neuropsychological assessment was conducted individually in a separate room on the second day after dialysis. Testing required approximately 90 min on average. A comprehensive battery of neuropsychological tests was designed to assess five cognitive domains: (1) attention using the Chinese modified version of the Trail Making Test A[12], Symbol Digit Modalities Test [15], (2) executive function using the Chinese modified version of the Trail Making Test B and A and modified version of the Stroop Color-Word Test [16]; (3) memory using the Chinese version of the Auditory Verbal Learning Test for short-delay and long-delay free recall [17] and Rey-Osterrieth Complex Figure (delayed recall test; Chinese version) [18]; (4) language using the Boston Naming Test (the 30-item version) and Category Verbal Fluency Test [19]; and (5) visuospatial function using the Rey-Osterrieth Complex Figure (copy test) [20].

Global cognition was assessed using the Chinese version of the MoCA-BJ and the Mini-Mental State Examination (MMSE) which is a 10 min screening test including questions to assess spatial and temporal orientation, immediate and delayed recall, language ability and oral command comprehension, and serial subtraction, and tasks of visuospatial ability [21, 22]. The MoCA-BJ requires educational adjustment; one point was added to the total score for those with ≤12 years' education [23]. We also evaluated depression using the Hamilton Depression Scale. Scores range from 0 to 63, with a score of ≥7 suggested as the optimal cutoff for suspected depression [24].

## Classification of cognitive impairment

We classified subjects as cognitive unimpaired or cognitive impairment based on criteria from the fifth version of the Diagnostic and Statistical Manual of Mental Disorders (DSM-V). Specifically, if the age- and education-adjusted scores were lower than1.5 SDs of the published norms in one test of one domain, the patient could be diagnosed as cognitive impairment [25]. The norms were based on a normative study of healthy, cognitively normal, community-dwelling adults in China [26].

## Ethics statement

The research project was done in accordance with the latest version of the Declaration of Helsinki and the study protocol was approved by the institutional ethical review board of Beijing Shijitan Hospital affiliated to Capital Medical University (approval no. SJT2016-18). All participants provided written informed consent before participating.

## Statistical analysis

All analyses were performed using SPSS software (version 19.0) (SPSS Inc, Chicago, IL, USA). The distribution of the data was analyzed by employing the Kolmogorov-Smirnov test. Data are presented as the mean ± SD for continuous variables with a normal distribution. Medians and interquartile ranges were used for continuous variables without a normal distribution and categorical variables.

To test the differences in the demographic and clinical characteristics and cognition scores between the cognitive impairment and cognitively unimpaired groups, normally distributed variables were compared using the Student's t-test, and non-normal distributed variables were compared using the Mann-Whitney U test. Categorical variables are presented as percentages and were compared using the Chi-square test.

To demonstrate the different impairment features of each cognitive domain, the raw scores for each neuropsychological test were T-transformed. The T-scores for each domain were then generated by averaging the T-scores of their respective tests; composite T-scores were computed by averaging the T-scores of the five cognitive domains, the Mann-Whitney U test was applied to compare the difference between cognitive impairment and cognitively unimpaired groups.

To examine the relationships between neurocognitive domains performance and MoCA and MMSE scores, a correlational analysis was performed using Spearman's rank correlation coefficient ($r_s$). To examine the validity between the total scores of MoCA-BJ and the test battery overall composite scores, a bivariate linear regression model was also calculated using the MoCA-BJ as the dependent variable and the cognitive composite score as the independent variable. The demographic variables including age, sex, and education were also added as covariates [13].

To further evaluate the discriminate predictive power of the MoCA-BJ and the MMSE against the DSM-V standards, a receiver operating characteristic (ROC) curve analysis was performed. The area under the curve (95% confidence interval, 95%CI), sensitivity, specificity, positive predictive value (PPV), and negative predictive value (NPV) of the MoCA-BJ and MMSE at various cutoffs were calculated. Optimal cutoff points were determined using the maximum value of Youden's index (calculated by sensitivity+ specificity-1) [9]. Statistical significance for all analyses was set at P value < 0.05.

# Results

## Demographic and clinical characteristics

Patients' demographic and clinical characteristics and the MoCA-BJ, and MMSE scores stratified by the cognitive function are presented in Table 1. Patients with cognitive impairment were more likely to be older and have a lower education level; a longer HD vintage; and comorbidities of diabetes, hypertension, and stroke. The MoCA-BJ score and MMSE score were significantly lower in the cognitive impairment group than the cognitively unimpaired group (Table 1).

## Characteristics of cognitive impairment

Of the 613 subjects, cognitive impairment was diagnosed in 496 (80.91%) using the DSM-V, and 117 (19.09%) were determined to have normal cognition. The T-scores in each cognitive domain in the cognitive impairment group were significantly lower than those in the cognitively unimpaired group (Table 2). Specifically, 75.69% (464/613) of patients were determined

**Table 1. Demographics and Clinical data for studied participants.**

| Characteristics | Total | Cognitively unimpaired | Cognitive impairment | P Value§ |
|---|---|---|---|---|
| | n = 613 | n = 117 | n = 496 | |
| Age (years) | 63.7±7.8 | 59.3±7.7 | 64.7±7.4 | < 0.001 |
| Female sex | 258(42.1%) | 47(40.2%) | 211(42.5%) | 0.641 |
| Education level | | | | 0.001 |
| ≤12 years | 444(72.4%) | 69(59.0%) | 375(75.6%) | |
| >12 years | 169(27.6%) | 48(41.0%) | 121(24.4%) | |
| Medical history | | | | |
| Smoking | 270(44.0%) | 51(43.6%) | 219(44.2%) | 0.925 |
| Alcohol intake | 261(42.6%) | 51(43.6%) | 210(42.3%) | 0.936 |
| Diabetes | 231(37.7%) | 33(28.2%) | 198(39.9%) | 0.019 |
| Hypertension | 545(88.9%) | 95(81.2%) | 450(90.7%) | 0.003 |
| Stroke | 100(16.3%) | 6(5.1%) | 94(19.0%) | <0.001 |
| CHD | 193(31.5%) | 36(30.8%) | 157(31.7%) | 0.853 |
| BMI (kg/m$^2$) | 23.6±4.1 | 24.2±5.8 | 23.5±3.6 | 0.202 |
| Dialysis vintage (months) | 57.0(24.0,101.5) | 42.0(12.0,78.0) | 60.0(29.0,104.8) | <0.001 |
| spKt/V | 1.5±0.5 | 1.4±0.3 | 1.5±0.5 | 0.246 |
| Lab. examination | | | | |
| Hb (g/L) | 111.1±14.6 | 110.7±15.3 | 111.2±14.5 | 0.737 |
| Alb (g/L) | 39.9±3.2 | 40.3±2.5 | 39.9±3.3 | 0.218 |
| Calcium (mmol/L) | 2.2±0.2 | 2.2±0.2 | 2.2±0.3 | 0.127 |
| Phosphate (mmol/L) | 1.7±0.7 | 1.8±0.7 | 1.7±0.6 | 0.342 |
| iPTH (pg/mL) | 187.6(103.5,358.2) | 167.6(107.0,311.5) | 201.8(103.4,372.2) | 0.058 |
| Depression scores | 4.0(1.0,7.0) | 4.0(2.0,7.0) | 3.0(1.0,7.0) | 0.441 |
| MoCA-BJ scores | 21.0(19.0,23.0) | 25.0(23.5,26.0) | 21.0(18.0,22.0) | <0.001 |
| MMSE scores | 25.0(23.0,27.0) | 27.0(26.0,29.0) | 25.0(23.0,26.0) | <0.001 |

Values are presented as the mean ± SD for normally distributed continuous variables, median (interquartile range) for non-normally distributed continuous variables, and n (%) for categorical variables.

§ Normally distributed variables were compared using the Student's t-test, and non-normal distributed variables were compared using the Mann-Whitney U test. Categorical variables are presented as percentages and were compared using the Chi-square test.

Abbreviations: CHD, coronary heart disease; BMI, body mass index; spKt/V, single-pool Kt/V; Hb, hemoglobin; Alb, albumin; iPTH, intact parathyroid hormone; MoCA, Montreal Cognitive Assessment; MMSE, Mini-Mental State Examination.

to have cognitive impairment according to the MoCA-BJ score, only 61.34% (376/613) of patients were determined to have CI according to the MMSE score.

## Associations between the MoCA-BJ, MMSE, and the standard neuropsychological battery of DSM-V

The Spearman's rank correlation analysis indicated that the MoCA-BJ was correlated with the composite scores of the five cognitive domains (rs = 0.639, p<0.001), specially in the domains of executive function ($r_s$ = 0.775, p<0.001), attention (rs = 0.564, p<0.001), visual space ($r_s$ = 0.457, p<0.001), memory ($r_s$ = 0.378, p<0.001) and language ($r_s$ = 0.237, p<0.001), whereas the MMSE had a weaker correlation with the standard neuropsychological tests (Table 3). The linear regression analysis revealed that the composite score explained 37.1% (β = 0.998, $R^2$ = 0.371, p<0.001) of the variance in the MoCA-BJ and the covariates including age (β = 0.18, $R^2$ = 0.062, p<0.05), sex (β = 0.12 $R^2$ = 0.051, p<0.05) and education (β = 0.23, $R^2$ = 0.084, p<0.05) only marginally affected variance in the performance.

**Table 2. T-scores of the standard neuropsychological assessments of five cognitive domains in the cognitively unimpaired and cognitive impairment groups.**

| Cognitive domains | Total | Cognitively impaired | Cognitive impairment | Z value | P value |
|---|---|---|---|---|---|
| | n = 613 | n = 117 | n = 496 | | |
| Global cognition | | | | | |
| MoCA | 21.0(19.0,24.0) | 26.0(24.0,27.0) | 21.0(18.0,23.0) | -13.389 | <0.001 |
| MMSE | 25.0(23.0,27.0) | 27.0(26.0,29.0) | 25.0(23.0,26.0) | -10.944 | <0.001 |
| Memory | 49.9(45.6,54.7) | 54.7(51.5,58.3) | 48.5(44.8,52.9) | -9.588 | <0.001 |
| AVLT 5 | 50.3(43.4,57.1) | 53.7(46.8,60.6) | 46.8(39.9,57.1) | -4.920 | <0.001 |
| AVLT1-5 | 49.0(43.1,56.5) | 53.2(48.1,59.0) | 47.3(40.8,55.7) | -5.805 | <0.001 |
| CFT- memory | 48.3(42.3,57.6) | 57.9(50.7,63.9) | 47.1(41.1,53.1) | -8.940 | <0.001 |
| Attention | 50.2(45.5,55.0) | 56.6(53.0,59.9) | 48.7(44.1,53.1) | -11.373 | <0.001 |
| SDMT | 52.3(46.6,56.1) | 56.6(53.4,60.6) | 50.2(44.2,54.9) | -9.679 | <0.001 |
| TMT-A | 49.4(44.6,55.1) | 57.6(52.7,62.0) | 47.0(42.1,52.7) | -11.427 | <0.001 |
| Executive function | 50.9(46.6,54.7) | 55.6(53.2,58.1) | 49.7(45.8,53.6) | -10.851 | <0.001 |
| TMT-B | 50.1(43.6,57.2) | 57.9(53.2,62.8) | 48.2(40.8,54.7) | -9.844 | <0.001 |
| SCWT-C | 53.3(46.7,56.0) | 54.7(52.0,57.3) | 53.3(45.3,56.0) | -5.526 | <0.001 |
| SCWT-T | 51.4(47.6,56.0) | 55.8(52.0,57.9) | 50.6(47.0,55.0) | -7.781 | <0.001 |
| Language | 50.4(45.9,54.8) | 53.9(50.4,58.6) | 49.7(45.2,53.9) | -6.736 | <0.001 |
| AFT | 49.7(42.3,55.4) | 55.4(47.9,61.0) | 47.9(42.3,55.4) | -6.412 | <0.001 |
| BNT | 54.8(48.2,58.1) | 54.8(51.5,58.1) | 51.5(44.9,54.8) | -3.608 | <0.001 |
| Visual space | 54.7(45.5,57.0) | 55.9(54.7,58.1) | 52.5(41.2,56.9) | -8.374 | <0.001 |
| CFT-copy | 54.7(45.5,57.0) | 55.9(54.7,58.1) | 52.5(41.2,56.9) | -8.374 | <0.001 |

Abbreviations: MoCA, Montreal Cognitive Assessment; MMSE, Mini-Mental State Examination; AVLT, Auditory-Verbal Learning Test; CFT, Complex Figure Test; SDMT, Symbol Digit Modalities Test; TMT, Trail Making Test; SCWT, Stroop Color, and Word Test; AFT, Animal verbal fluency test; BNT, Boston Naming Test

## ROC curve analysis

The ROC analysis identified an optimal cut-off for the MoCA at ≤ 24 points (Table 4), with a sensitivity of 87.7% (95% CI, 0.851–0.903), a specificity of 75.2% (95% CI, 0.716–0.788) and an area under the curve (AUC) of 0.891 (95% CI, 0.859–0.924). In comparison, the MMSE only had a sensitivity of 70.4% (95% CI, 0.668–0.740), and a specificity of 76.9% (95% CI, 0.736–0.802, and AUC of 0.823 (95% CI, 0.786–0.860) for an optimal cut-off of ≤ 26 points (Table 4, Fig 1).

**Table 3. The correlations of the Beijing version of the Montreal Cognitive Assessment and the Mini-Mental State Examination with composite T-scores and each cognitive domain.**

| Cognitive domains | MoCA | | MMSE | |
|---|---|---|---|---|
| | $r_s$ | P-value | $r_s$ | P-value |
| Memory | 0.378 | <0.001 | 0.192 | <0.001 |
| Attention | 0.564 | <0.001 | 0.272 | <0.001 |
| Executive function | 0.775 | <0.001 | 0.353 | <0.001 |
| Language | 0.237 | <0.001 | 0.043 | 0.288 |
| Visual space | 0.457 | <0.001 | 0.211 | <0.001 |
| Composite | 0.639 | <0.001 | 0.296 | <0.001 |

Abbreviations: MoCA, Montreal Cognitive Assessment; MMSE, Mini-Mental State Examination

# Discussion

In this multicenter observational study, we validated that the MoCA-BJ was an effective and sensitive screening tool for the early detection of cognitive impairment identified by the DSM-V in patients undergoing HD. Furthermore, the refined cutoff point of 24 for the MoCA-BJ could provide optimal sensitivity and specificity in screening for cognitive impairment in these patients. However, the MMSE only had a lower power of discrimination of cognitive impairment in hemodialysis patients.

Previous validation studies have evaluated the efficacy of the MoCA in screening cognitive impairment in a variety of populations[9, 27, 28], although the original data collected in a group of patients with AD and mild cognitive impairment (MCI) showed the optimal cut-off value should be less than 26[20], several international validations showed wide differences in thresholds, sensitivity and specificity values [29]. Damian AM et al [30] evaluated the predictive value of the MoCA in the detection of cognitive impairment in 135 American subjects, the results showed that the MoCA threshold of 26 appeared to be optimal in primary care. Luis CA et al [31] also found that Using the recommended cut-off score of 26, the MoCA detected 97% of those with cognitive impairment but specificity was fair (35%). Using a lower cut-off score of 23, the MoCA exhibited excellent sensitivity (96%) and specificity (95%) in 118 English-speaking older American adults. All these validation analyses remind us to make further examination about the application of the MoCA in different kinds of patients. However, the reliability and validity of the MoCA in patients undergoing HD have not been fully explored [32, 33]. Recently, Tiffin-Richards FE et al [13] demonstrated that the MoCA had good levels of sensitivity and specificity for screening cognitive impairment in a moderately sized sample of HD patients. In another validation study, Lee SH et al [34] compared the Korean version of the MoCA to the MMSE in 30 patients undergoing HD with a matched control group; the results indicated that the Korean version of the MoCA seemed to be more sensitive than the MMSE in HD patients. Our results revealed that the MoCA-BJ was more sensitive and had more predictive power than the MMSE in the detection of mild CI in the Chinese HD population (Table 4, Fig 1), which was consistent with the findings of the abovementioned studies and validated these results in a large group of patients undergoing HD. This is also the first validation analysis of MoCA-BJ among Chinese patients undergoing HD.

**Table 4. Discriminant validity of the Beijing version of the Montreal Cognitive Assessment and the Mini-Mental State Examination for detecting cognitive impairment.**

| MoCA | | | | | MMSE | | | | |
|---|---|---|---|---|---|---|---|---|---|
| Cutoff ≤ | Sen | Spe | PPV | NPV | Cutoff ≤ | Sen | Spe | PPV | NPV |
| 20 | 0.345 | 0.983 | 0.988 | 0.261 | 20 | 0.101 | 1.000 | 1.000 | 0.208 |
| 21 | 0.492 | 0.983 | 0.992 | 0.313 | 21 | 0.111 | 1.000 | 1.000 | 0.210 |
| 22 | 0.657 | 0.880 | 0.959 | 0.377 | 22 | 0.169 | 1.000 | 1.000 | 0.221 |
| 23 | 0.806 | 0.803 | 0.946 | 0.495 | 23 | 0.234 | 1.00 | 1.000 | 0.235 |
| 24† | 0.877 | 0.752 | 0.938 | 0.591 | 24 | 0.371 | 1.000 | 1.000 | 0.273 |
| 25 | 0.942 | 0.598 | 0.909 | 0.707 | 25 | 0.498 | 0.897 | 0.954 | 0.297 |
| 26 | 0.988 | 0.393 | 0.873 | 0.885 | 26† | 0.704 | 0.769 | 0.928 | 0.380 |
| 27 | 1.000 | 0.222 | 0.845 | 1.000 | 27 | 0.792 | 0.701 | 0.918 | 0.443 |
| 28 | 1.000 | 0.103 | 0.825 | 1.000 | 28 | 0.869 | 0.470 | 0.874 | 0.458 |
| 29 | 1.000 | 0.103 | 0.825 | 1.000 | 29 | 0.946 | 0.299 | 0.851 | 0.565 |

Abbreviations: MoCA, Montreal Cognitive Assessment; MMSE, Mini-Mental State Examination; Sen, Sensitivity; Spe, Specificity; PPV, positive predictive value; NPV, negative predictive value.

† Optimal cutoff score determined using Youden's index.

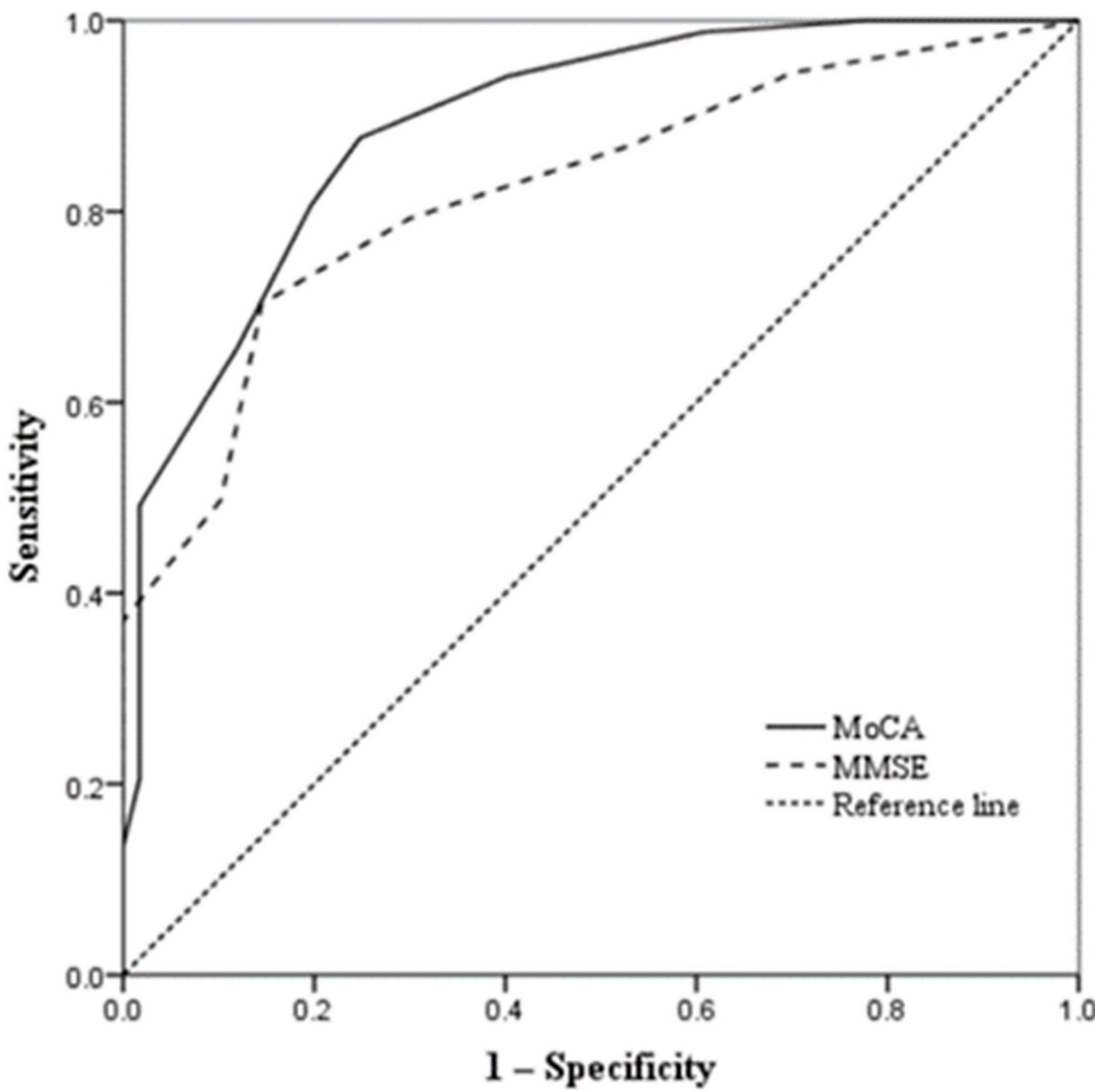

**Fig 1. Receiver operating characteristic curve analysis of the Mini-Mental State Examination and the Montreal Cognitive Assessment.**

Exploring the reason why the MoCA-BJ is superior to MMSE in patients undergoing HD, the features of cognitive impairment in patients undergoing HD and the contents of the MoCA-BJ might be related. First, both those of previous studies and our data indicated that cognitive impairment in patients undergoing HD was more related to the injuries in the domains of executive function and visuospatial abilities [35]. In addition, patients undergoing HD often have a high prevalence of cerebrovascular risk factors, such as older age, hypertension, diabetes, dyslipidemia, high total plasma homocysteine level, oxidant stress, and inflammation [36, 37]. Secondly, the MoCA-BJ incorporated subtests assessing executive functions and psychomotor speed, which are frequently impaired in vascular cognitive impairment. In

these regards, the focus of the MoCA on executive functions was believed to be one of the main reasons why it was superior to the MMSE in HD populations. The MMSE was designed for the detection of cognitive deficits observed in the process of dementia, particularly regarding attention, memory, and language functions. It lacks the ability to assess executive domains; thus, its sensitivity for detecting cognitive impairment in patients undergoing HD is lower than that of the MoCA.

Another important result of our study was that the optimal MoCA-BJ cutoff score was 24, as determined by Youden's index, which was lower than the commonly recommended cutoff score of 26 in the general population. Although the general MoCA-BJ cutoff ($\leq$26) had excellent sensitivity (0.942), the specificity was poor (0.598) in this group of patients undergoing HD. When we applied the optimal cutoff (i.e., $\leq$24), the specificity improved to 0.752, which was nearly equivalent to that of the MMSE (0.795), but the sensitivity was reduced to 0.877, although this was still higher than that of the MMSE (0.704). Several other studies have determined lower values in different populations e.g., a cutoff of 23.5 in a population with mild cognitive impairment [38] and of 21/22 in a population with cerebral small vessel disease [39]. The possible reason for this difference might be attributed to the difference between the original MoCA study and our study about age, race, education level, clinical history, and the diagnostic criteria for cognitive impairment. It seems that the optimal cutoff value should be applied when the MoCA-BJ is used to assess Chinese patients undergoing HD.

Some strengths of our study include the detailed neuropsychological testing with published norms on all tests in all cognitive domains, and the diagnosis of cognitive impairment based on the criteria of the DSM-V, which enabled a comprehensive and standard evaluation of cognition and the interpretation of concurrent validity. In addition, to improve the quality and reliability of our study, all neuropsychological tests were performed by research staff, who were trained centrally by the study neuropsychologist and received certificates before the study commencement. However, there were still several limitations associated with our study. Given that the present data were obtained from Chinese patients undergoing HD in Beijing, these results may not be generalizable to other areas in China, because there is much geographic and cultural variability across China, and specific regional characteristics may influence the performance of the residents on the neuropsychological tests. Future studies should include patients undergoing HD from more regions of the country.

In conclusion, the MoCA-BJ appears to be a good sensitivity and specificity levels in detecting cognitive impairment in hemodialysis patients. Our data support that the MoCA-BJ can be applied for the screening of cognitive impairment in routine practice in these patients.

## Supporting information

**S1 File. STROBE statement.**
(DOCX)

**S1 Table. Basic characteristics of the participants.**
(XLSX)

**S2 Table. Neuropsychological tests of five cognitive domains of the participant.**
(XLSX)

## Acknowledgments

We would like to thank the research and clinical unit staff at different hemodialysis centers of 11 hospitals in Beijing for their support and collaboration in this research.

## Author Contributions

**Conceptualization:** Yang Luo.

**Data curation:** Ru Tian, Chunxia Zhang.

**Formal analysis:** Ru Tian.

**Methodology:** Yidan Guo, Pengpeng Ye, Chunxia Zhang.

**Software:** Pengpeng Ye.

**Supervision:** Pengpeng Ye, Yang Luo.

**Writing – original draft:** Ru Tian.

**Writing – review & editing:** Yidan Guo.

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
