## [Decision Letter · Decision Letter 0]

30 Oct 2019

PONE-D-19-21883

The validation of the Beijing version of the Montreal Cognitive Assessment in Chinese patients undergoing hemodialysis

PLOS ONE

Dear Dr. Luo,

Thank you for submitting your manuscript to PLOS ONE. After careful consideration, we feel that it has merit but does not fully meet PLOS ONE’s publication criteria as it currently stands. Therefore, we invite you to submit a revised version of the manuscript that addresses the points raised during the review process.

We would appreciate receiving your revised manuscript by Dec 14 2019 11:59PM. To enhance the reproducibility of your results, we recommend that if applicable you deposit your laboratory protocols in protocols.io, where a protocol can be assigned its own identifier (DOI) such that it can be cited independently in the future. For instructions see: http://journals.plos.org/plosone/s/submission-guidelines#loc-laboratory-protocols

We look forward to receiving your revised manuscript.

Kind regards,

Jeannie-Marie Sheppard Leoutsakos, PhD

Academic Editor

PLOS ONE

Journal Requirements:

3. There are several cases where P-values are reported as being equal to 0. Please correct and clarify.

Reviewers' comments:

Reviewer's Responses to Questions

**Comments to the Author**

1. Is the manuscript technically sound, and do the data support the conclusions?

Reviewer #1: Partly

Reviewer #2: Yes

2. Has the statistical analysis been performed appropriately and rigorously? 

Reviewer #1: Yes

Reviewer #2: Yes

3. Have the authors made all data underlying the findings in their manuscript fully available?

Reviewer #1: Yes

Reviewer #2: Yes

4. Is the manuscript presented in an intelligible fashion and written in standard English?

Reviewer #1: No

Reviewer #2: No

5. Review Comments to the Author

Reviewer #1: The ms. is focused on a contribution to the validation of the Beijing version of the MoCA (MoCA-BJ) for detecting Cognitive Impairments among a group of haemodialysis patients.

I have some concerns with the ms. In particular:

• Review the Abstract

Please revise the English

• Review the Introduction:

I suggest to improve this section, it is very gaunt, providing scarce information. First of all, it is not clear which kind of cognitive impairment the authors are talking about. Moreover, before starting to investigate the prevalence of the disorder in the hemodialysis population, it’d be good to do a general overview of the disorder, also in other populations, citing recent articles published on the same topic such as Bosco et al., 2017; Siciliano et al., 2019; Huang et al., 2018.

• Review the Subjects and Methods section:

Rename the section “method”;

Move the subsection “ethics statement” before the “statistical analysis”;

Rename the subsection “Study participants” in “participants”. Add in the text some demographic information such as means and standard deviation about age and level of education, and gender composition, as well. the paragraph needs to be rewrite taking into account English grammar and syntactic rules.

Au.s should clarify in the subsection “Classification of cognitive function” what they intend for “cognitive impairment”, specifying better also the criteria of DSM-5 used to diagnose the impairment. Reading is not easy.

Add references after Roc Curves and Youden Index in the statistical analysis section.

• Review the Results section and Discussion

In table 1 substitute the label p-value with Test

Add references after this sentence: “Previous studies have evaluated the efficacy of the MoCA and the MMSE in screening cognitive impairment in a variety of populations”

It is necessary to check some typos and errors, and, again, to revise the English in order to make the ms. easily legible.

• Notwithstanding the scarce literature about haemodialysis patients affected by “cognitive impairment” it is necessary to integrate the bibliographical apparatus with more recent research.

Reviewer #2: This study details the use of the MoCA-Beijing (BJ) in hemodialysis patients. The authors report on the MoCA-BJ’s correlation with the MMSE as well as with neuropsychological tests. Diagnostic accuracy of the MoCA-BJ is also assessed and the authors find that the optimal cutoff for cognitive impairment is 24, which is consistent with several previously published studies of the English version of the MoCA. Below are my comments.

General Comments

1. Although the English is generally good, there are several instances where the text is grammatically incorrect.

2. Throughout the manuscript, the authors refer to their subjects as cognitive impairment and non-cognitive impairment. The preferred term for non-cognitive impairment is “cognitively unimpaired” (Jack CR, et al. NIA-AA Research Framework: toward a biological definition of Alzheimer’s disease. Alzheimers Dementia. 2018;14.4:535–62.)

Statistical Analysis

1. When describing the analysis of categorical variables, please spell out Chi-square and only use the Greek symbol for Chi when reporting the results from statistical tests.

Results

1. In the bivariate linear regression model, were any demographic variables such as age, sex, and education adjusted for? This is particularly relevant on page 13 where the authors report the amount of variance explained. Perhaps they could report R-squared values from adjusted and unadjusted models.

2. Throughout the text and in the tables, p-values of 0.000 are reported. Although this is the default output for small p-values in SPSS, this is not the way they should be reported. For all cases where the p-value is 0.000, please replace with <0.001.

3. For the measures of diagnostic accuracy, the authors report 95% CIs for the AUC values but not for sensitivity and specificity. 95% CIs for sensitivity and specificity should be reported as well. If these are not given by SPSS, the equation for their calculation is relatively simple and is easily available on a number of reputable websites.

Discussion

1. The authors’ finding that 24 is the optimal cutoff for the identifying cognitive impairment is similar to other findings and these studies should be cited and briefly discussed:

Damian, A.M., Jacobson, S.A., Hentz, J.G., Belden, C.M., Shill, H.A., Sabbagh, M.N., et. al (2011). The Montreal Cognitive Assessment and the Mini-Mental State Examination as screening instruments for cognitive impairment: Item analyses and threshold scores. Dementia and Geriatric Cognitive Disorders, 31, 126-131.

Coen, R.F., Cahill, R., & Lawlor, B.A. (2011). Things to watch out for when using the Montreal Cognitive Assessment (MoCA). International Journal of Geriatric Psychiatry, 26, 106- 108.

Luis CA, Keegan PA, & Mullan M. (2009). Cross validation of the Montreal Cognitive

Assessment in community dwelling older adults residing in the Southeastern US. International Journal of Geriatric Psychiatry, 24, 197-201.

Gluhm, S., Goldstein, J., Loc, K., Colt, A., Van Liew, C., & Corey-Bloom, J. (2013). Cognitive performance on the Mini-Mental State Examination and the Montreal Cognitive Assessment across the healthy adult lifespan. Cognitive and Behavioral Neurology, 26, 1-5.

6. PLOS authors have the option to publish the peer review history of their article (what does this mean?). If published, this will include your full peer review and any attached files.

Reviewer #1: Yes: Prof. Andrea Bosco PhD

Reviewer #2: No

---

## [Author Response · Author response to Decision Letter 0]

23 Nov 2019

Nov 22th 2019

PONE-D-19-21883 (The validation of the Beijing version of the Montreal Cognitive Assessment in Chinese patients undergoing hemodialysis)

PLOS ONE 

Dear Editor Jeannie-Marie Sheppard Leoutsakos

Here is the response letter from the authors of PONE-D-19-21883, we would like to express our appreciation to you and reviewers for your kind work and valuable suggestions. Now we have read the comments carefully and revised our manuscript accordingly. Here below are some of answers to the questions of the reviewers. 

At the same time, we understand that our language ability is limited, so we have sent our manuscript to the editage for language editing, and we have uploaded the certificate of that editing to you. We also uploaded the anonymized data set of our study as supplementary documents for replicating our study founding. 

If there are still other limitations, please let us know, and we are looking forward to hearing from you soon.

Sincerely

Yang Luo.

Reviewer #1: The ms. is focused on a contribution to the validation of the Beijing version of the MoCA (MoCA-BJ) for detecting Cognitive Impairments among a group of haemodialysis patients.

I have some concerns with the ms. In particular:

• Review the Abstract

Please revise the English

We have asked the Editage, one of renown English language editing workshops, to improve our expression in the manuscript. We have uploaded the certificate of the editing by the Editage as a supplementary document. 

• Review the Introduction:

I suggest to improve this section, it is very gaunt, providing scarce information. First of all, it is not clear which kind of cognitive impairment the authors are talking about. Moreover, before starting to investigate the prevalence of the disorder in the hemodialysis population, it’d be good to do a general overview of the disorder, also in other populations, citing recent articles published on the same topic such as Bosco et al., 2017; Siciliano et al., 2019; Huang et al., 2018.

We fully agree with your opinion that the cognitive impairment in the hemodialysis patient includes mild and major cognitive impairment. We have already learned the papers that you recommended to us and try our best to add information about the related cognitive impairment in other populations in our introduction.

• Review the Subjects and Methods section:

Rename the section “method”; we have changed accordingly.

Move the subsection “ethics statement” before the “statistical analysis”. We have moved the ethics statement to the right place.

Rename the subsection “Study participants” in “participants”. Add in the text some demographic information such as means and standard deviation about age and level of education, and gender composition, as well. the paragraph needs to be rewrite taking into account English grammar and syntactic rules. We have renamed this subsection as “participants” under your suggestion, and provide some demographic information. 

Au.s should clarify in the subsection “Classification of cognitive function” what they intend for “cognitive impairment”, specifying better also the criteria of DSM-5 used to diagnose the impairment. Reading is not easy. We have revised this section, try to make it clear to express the diagnosis criteria of cognitive impairment.

Add references after Roc Curves and Youden Index in the statistical analysis section. We have added references in this section.

• Review the Results section and Discussion

In table 1 substitute the label p-value with Test. We have added notes in Table 1 to indicated the tests used in the analysis.

Add references after this sentence: “Previous studies have evaluated the efficacy of the MoCA and the MMSE in screening cognitive impairment in a variety of populations”. We have added related references and introduced some of the important results from previous studies.

It is necessary to check some typos and errors, and, again, to revise the English in order to make the ms. easily legible. As we have mentioned above, we provided the certificate of language editing from the Editage.

• Notwithstanding the scarce literature about haemodialysis patients affected by “cognitive impairment” it is necessary to integrate the bibliographical apparatus with more recent research. We agree with your suggestion and searched the related papers from Pubmed and revised our manuscript accordingly. 

Reviewer #2: This study details the use of the MoCA-Beijing (BJ) in hemodialysis patients. The authors report on the MoCA-BJ’s correlation with the MMSE as well as with neuropsychological tests. Diagnostic accuracy of the MoCA-BJ is also assessed and the authors find that the optimal cutoff for cognitive impairment is 24, which is consistent with several previously published studies of the English version of the MoCA. Below are my comments.

General Comments

1. Although the English is generally good, there are several instances where the text is grammatically incorrect. Thank you for the suggestions. In order to improve the quality of our manuscript, we have asked the Editage, one of renown English language editing workshops, to improve our expression in the manuscript. We have uploaded the certificate of the editing by the Editage as supplementary document. 

2. Throughout the manuscript, the authors refer to their subjects as cognitive impairment and non-cognitive impairment. The preferred term for non-cognitive impairment is “cognitively unimpaired” (Jack CR, et al. NIA-AA Research Framework: toward a biological definition of Alzheimer’s disease. Alzheimers Dementia. 2018;14.4: 535–62.)

We have read the article and changed the old terms in our manuscript, thank you for your suggestion.

Statistical Analysis

1. When describing the analysis of categorical variables, please spell out Chi-square and only use the Greek symbol for Chi when reporting the results from statistical tests.

We have changed Greek symbols into normal spelling words, thank you again.

Results

1. In the bivariate linear regression model, were any demographic variables such as age, sex, and education adjusted for? This is particularly relevant on page 13 where the authors report the amount of variance explained. Perhaps they could report R-squared values from adjusted and unadjusted models.

After reading some references, we understood the importance to add some demographic variables like age, sex, and education in our linear regression analysis, we have provided the related R-squared in the related section of our results.

2. Throughout the text and in the tables, p-values of 0.000 are reported. Although this is the default output for small p-values in SPSS, this is not the way they should be reported. For all cases where the p-value is 0.000, please replace with <0.001.

We have replaced the old expression and changed in the style as you suggested.

3. For the measures of diagnostic accuracy, the authors report 95% CIs for the AUC values but not for sensitivity and specificity. 95% CIs for sensitivity and specificity should be reported as well. If these are not given by SPSS, the equation for their calculation is relatively simple and is easily available on a number of reputable websites.

We have added the 95% CIs for the sensitivity and specificity in the results of ROC analysis.

Discussion

1. The authors’ finding that 24 is the optimal cutoff for the identifying cognitive impairment is similar to other findings and these studies should be cited and briefly discussed:

We have downloaded these papers and cited them in our discussion. The contents of these papers provide a lot of important information in this area, we would like to thank you again. 

Damian, A.M., Jacobson, S.A., Hentz, J.G., Belden, C.M., Shill, H.A., Sabbagh, M.N., et. al (2011). The Montreal Cognitive Assessment and the Mini-Mental State Examination as screening instruments for cognitive impairment: Item analyses and threshold scores. Dementia and Geriatric Cognitive Disorders, 31, 126-131.

Coen, R.F., Cahill, R., & Lawlor, B.A. (2011). Things to watch out for when using the Montreal Cognitive Assessment (MoCA). International Journal of Geriatric Psychiatry, 26, 106- 108.

Luis CA, Keegan PA, & Mullan M. (2009). Cross validation of the Montreal Cognitive

Assessment in community dwelling older adults residing in the Southeastern US. International Journal of Geriatric Psychiatry, 24, 197-201.

Gluhm, S., Goldstein, J., Loc, K., Colt, A., Van Liew, C., & Corey-Bloom, J. (2013). Cognitive performance on the Mini-Mental State Examination and the Montreal Cognitive Assessment across the healthy adult lifespan. Cognitive and Behavioral Neurology, 26, 1-5.

---

## [Decision Letter · Decision Letter 1]

12 Dec 2019

The validation of the Beijing version of the Montreal Cognitive Assessment in Chinese patients undergoing hemodialysis

PONE-D-19-21883R1

Dear Dr. Luo,

We are pleased to inform you that your manuscript has been judged scientifically suitable for publication and will be formally accepted for publication once it complies with all outstanding technical requirements.

With kind regards,

Jeannie-Marie Sheppard Leoutsakos, PhD

Academic Editor

PLOS ONE

Additional Editor Comments (optional):

Reviewers' comments:

Reviewer's Responses to Questions

**Comments to the Author**

1. If the authors have adequately addressed your comments raised in a previous round of review and you feel that this manuscript is now acceptable for publication, you may indicate that here to bypass the “Comments to the Author” section, enter your conflict of interest statement in the “Confidential to Editor” section, and submit your "Accept" recommendation.

Reviewer #1: All comments have been addressed

Reviewer #2: All comments have been addressed

2. Is the manuscript technically sound, and do the data support the conclusions?

Reviewer #1: Yes

Reviewer #2: Yes

3. Has the statistical analysis been performed appropriately and rigorously? 

Reviewer #1: Yes

Reviewer #2: Yes

4. Have the authors made all data underlying the findings in their manuscript fully available?

Reviewer #1: No

Reviewer #2: Yes

5. Is the manuscript presented in an intelligible fashion and written in standard English?

Reviewer #1: Yes

Reviewer #2: Yes

6. Review Comments to the Author

Reviewer #1: It seems to me that authors have made all the requested revisions.

Minor concers

In the list of references I found after some authors' names, the fragment "AUID- Oho", please identify it and delete.

Reviewer #2: The authors have responded sufficiently to my comments. No additional revisions are needed for this manuscript.

7. PLOS authors have the option to publish the peer review history of their article (what does this mean?). If published, this will include your full peer review and any attached files.

Reviewer #1: Yes: Prof. Andrea Bosco

Reviewer #2: Yes: Michael Malek-Ahmadi

---

## [Editor Report · Acceptance letter]

19 Dec 2019

PONE-D-19-21883R1 

The validation of the Beijing version of the Montreal Cognitive Assessment in Chinese patients undergoing hemodialysis 

Dear Dr. Luo:

I am pleased to inform you that your manuscript has been deemed suitable for publication in PLOS ONE. Congratulations! Your manuscript is now with our production department. 

With kind regards,

on behalf of

Dr. Jeannie-Marie Sheppard Leoutsakos 

Academic Editor

PLOS ONE